# OpenReview forum: "Stoichiometry Representation Learning with Polymorphic Crystal Structures"
_ICLR.cc/2024/Conference — Submitted to ICLR 2024_

### Official Review · Reviewer_jsDD · 2023-10-27

**Soundness:** 3 good
**Presentation:** 3 good
**Contribution:** 2 fair
**Rating:** 6
**Confidence:** 2

**Summary:**

This paper proposes PolySRL, which is a combination of previous self-supervise learning technics, to the task of crystal property preidction using structural and compositional information during training and only compositional information during inference. They show that instead of using deterministic way to obtain representations from crystal compositional information, it is better to use a distribution and a sampling process, due to the one-to-many nature between compositional information and structures.

**Strengths:**

- The first work to discuss using self supervise learning technics for crystal property prediction tasks without structural information.
- Clear writing, with informative figures and extensive experiments

**Weaknesses:**

- The novelty is limited, due to the fact that the proposed pipeline is a combination of previous technics.
- Improvements beyond baselines without structural information during training is not significant but in the near range. The errors are not very small and I'm a little concerned how practically useful the predictions will be.
- The global idea of predicting properties based on crystal compositional information is a little bit tricky, due to that crystal properties are largely determined by the structure. As you mentioned in your figure 1, diamond and graphite have totally different properties.

**I have viewed the rebuttals from authors.**

Thank you for providing detailed information in the Appendix H and in your rebuttals. **I read into details in your responses and now my concern about predicting a property value with an uncertainty value is addressed.**

It is addressed because, when filtering potential compositions of crystals for downstream tasks, one can use the property prediction together with the uncertainty value to have an approximation property upper and lower bound, e.g., (prediction - uncertainty, prediction + uncertainty). And I tend to believe this is of good value and potential to decrease the potential exploration space of compositions.

Since this is the major concern I had when I gave the score of 5, I tend to increase the score to 6, with confidence 2. **The final recommendation will be, this is a borderline paper with an interesting proposal, however, more experiments about the usage of uncertainty are needed, and the novelty in terms of methodology is a little limited.**

**Questions:**

As I mentioned in the weaknesses.

---

> ### Author Response · Authors · 2023-11-14
>
> Thank you for recognizing that our work is pioneering work for crystal property prediction tasks without structural information! We are more than willing to clarify the reviewer’s concerns one by one.
>
> **W1. Limited Novelty.**
>
> As the reviewer pointed out, this paper is the first work that utilizes self-supervised learning to learn the representation of stoichiometry for various downstream tasks. While techniques in our paper are highly inspired by previous works, we believe that our contributions come from 1) identifying a novel and important problem, 2) suggesting a suitable method for the problem, and 3) providing insightful results that can progress the field with the method.
>
> **Novelty and Importance of the Problem.**
> Recently, the majority of machine learning (ML) models in materials science are built on top of the crystal structure that are computed based Density functional Theory (DFT). However, in real-world wet lab experiments, it is common situation where we only have compositional information (i.e., stoichiometry) without structural information as also mentioned by [reviewer rRHK](https://openreview.net/forum?id=yuGqe96t9O&noteId=B4csbUBOsR), due to the atomic-level rearrangement of the materials during the materials synthesis process (We have provided a more detailed explanation on this situation by addressing [W3](https://openreview.net/forum?id=yuGqe96t9O&noteId=efaOs4V5j0) in the following) [1, 2]. This missing structural information complicates the application of such structure-based ML models in the real-world materials discovery process. Therefore, one promising avenue for enhancing the practicality of ML in materials discovery lies in the prediction of materials property without structural information, i.e. relying solely on stoichiometry. However, as the reviewer mentioned in [W3](https://openreview.net/forum?id=yuGqe96t9O&noteId=efaOs4V5j0), it is also widely known that crystal structure heavily determines the properties of the materials.
>
>
> **Suitability of the Method.**
> As a solution, we proposed PolySRL, which distills the knowledge of DFT-calculated structure into the stoichiometry encoders during the pre-training stage, and fine-tune or evaluate the model solely on stoichiometry. One straightforward approach would be contrasting the object with its 3D representation, which has shown promising results in organic molecules [3, 4]. However, our extensive experiments reveal that these methods do not adequately capture the complexities of inorganic materials, such as polymorphism. To this end, we propose to learn a probabilistic representation that can learn not only the appropriate representation of stoichiometry but uncertainty induced by polymorphism.
>
> **Insightful Results.** Thanks to the suitability of methods for the task, we could make several insightful observations from experimental results. Briefly, uncertainty provided by models contains the information of stoichiometry regarding 1) a number of polymorphic structures, 2) computational challenges in calculating the accurate structure, and 3) similarity between polymorphic structures, all of which are common interest in materials science.
>
> In conclusion, we fully agree with the reviewer that innovations in algorithms are crucial for the field's advancement. Concurrently, we also argue that a comprehensive investigation beyond novel algorithms, such as providing novel problems and insightful results, is also essential for advancing the field. Hence, we believe that PolySRL has contributed to the field's development by introducing problems that are more relevant to real-world applications, as well as by delivering meaningful results that have been long common interest in materials science.
>
> [1] Predicting the band gaps of inorganic solids by machine learning, The journal of physical chemistry letters 2018.
>
> [2] Predicting materials properties without crystal structure: Deep representation learning from stoichiometry, Nature communications 2020.
>
> [3] 3d infomax improves gnns for molecular property prediction, ICML 2022.
>
> [4] Pre-training molecular graph representation with 3d geometry, ICLR 2023.

---

> ### Author Response · Authors · 2023-11-14
>
> **W2-1. Statistically significant improvement.**
>
> To exhibit the statistical significance of the enhancement, we conducted a paired t-test within the representation learning scenario for each dataset. To do so, we additionally conducted 5 separate instances of 5-fold cross-validation, with each instance featuring entirely distinct data splits (amounting to a total of 25 individual runs), and then comparing the means from each set of cross-validation results. The resulting numbers in table represent the p-values from the paired t-tests conducted with PolySRL under the following hypotheses:
>
> $H_0 : \mu_{PolySRL} = \mu_{Baseline}$
>
> $H_1 : \mu_{PolySRL} > \mu_{Baseline}$
>
> |             | Band G.  | Form. E. | Metallic | E.C. (300K) | T.C. (300K) | Seebeck (300K) | E.C. (600K) | T.C. (600K) | Seebeck (600K) |
> |-------------|----------|----------|----------|-------------|-------------|----------------|-------------|-------------|----------------|
> | GraphCL     | 3.80E-04 | 4.18E-03 | 7.63E-03 | 3.12E-03    | 4.29E-04    | 2.38E-03       | 6.72E-04    | 1.39E-02    | 4.07E-02       |
> | MP Band G.  | 2.48E-01 | 1.20E-04 | 6.53E-04 | 4.48E-04    | 2.15E-04    | 4.55E-04       | 5.90E-04    | 1.47E-03    | 1.40E-03       |
> | MP Form. E. | 5.14E-02 | 9.23E-01 | 1.89E-02 | 3.93E-04    | 1.75E-03    | 5.03E-03       | 1.47E-03    | 3.23E-03    | 4.30E-02       |
> | 3D Infomax  | 1.21E-03 | 1.40E-03 | 3.42E-02 | 1.44E-02    | 2.72E-02    | 8.39E-04       | 6.06E-03    | 3.89E-02    | 5.64E-02       |
>
> We have the following observations:
> 1) PolySRL achieves statistically significant improvements over baseline methods that do not leverage structural information (such as GraphCL, MP Band G., MP Form. E.) at the 0.05 significance level except for Band G. and Form. E. datasets.
> 2) On the other hand, in the datasets for Band G. and Form. E., PolySRL does not achieve statistical significance compared to models that benefit from the corresponding DFT-calculated values. For example, in Band G. dataset, we observe that the model trained with DFT-calculated band gap (i.e., MP Band G.) outperforms PolySRL, which was also described in observation 3 in Section 5.2. in the main paper. However, as discussed in the main text, these models are specifically tailored for particular properties, which may limit their broader applicability across a range of tasks.
>
> **W2-2. Practical usage of PolySRL.**
>
> Moreover, in terms of practical application, a further appeal of our model lies in its provision of uncertainty metrics. The quantification of uncertainties in model predictions is a critical factor when considering the deployment of machine learning in real-world scenarios, particularly for virtual screening in the scientific discovery process [1, 2]. Furthermore, comprehensive experiments on uncertainty presented in Section 5 show that the uncertainties produced by PolySRL are in consistent agreement with the established domain knowledge in materials science. Consequently, we argue that our model possesses several qualities that render it highly beneficial for practical use in the field.
>
> [1] Machine learning framework for enhanced molecular screening. Chemical science 2021.
>
> [2] Accelerating high-throughput virtual screening through molecular pool-based active learning. Chemical science 2021.

---

> ### Author Response · Authors · 2023-11-14
>
> **W3. The global idea of predicting properties solely based on compositional information is tricky.**
>
> First of all, we appreciate the reviewer for raising an important discussion point! We fully understand the reviewer's comment that this topic is tricky, since most machine learning methods rely on structural data of crystalline materials. To further clarify the paper's contributions, we have added this topic to the discussion section in **Appendix H**.
>
> As the reviewer mentioned, crystal property is highly determined by the structural information of the crystal. Most of the machine learning approaches for crystal property prediction utilize crystal structure information as an input, which is obtained through Density Functional Theory (DFT) calculations. However, most of the real-world wet lab experimental scenarios lack suitable structural information on the crystal due to the uncertainty of atomic arrangements [1, 2].
>
> More specifically, during the synthesis process of materials, atomic-level rearrangement of the material occurs through the mixing of raw materials, heat treatment, and solvent reactions. Therefore, even if the crystal structure of the raw material is known, the synthesized material may have a new crystal structure due to thermodynamic uncertainty introduced by the synthesis operation, making it impossible to determine the crystal structure precisely.
>
> Various chemical analysis techniques, such as X-ray diffraction (XRD) [3], have been developed to identify the crystal structure of synthesized materials, but due to cost and limitations in analytical accuracy, the crystal structure of synthesized materials is not typically the focus of analysis in actual chemical experiments.
>
> In conclusion, a lack of knowledge about the crystal structure is a typical scenario in real-world wet laboratory experiments. Consequently, we argue that predicting properties using only compositional information aligns more closely with practical applications in the real-world material discovery process.
>
> [1] Predicting the band gaps of inorganic solids by machine learning, The journal of physical chemistry letters 2018.
>
> [2] Predicting materials properties without crystal structure: Deep representation learning from stoichiometry, Nature communications 2020.
>
> [3] X-ray diffraction (XRD) techniques for materials characterization, Materials characterization using nondestructive evaluation (NDE) methods 2016.
>
> **Final Remark.**
>
> Again, thank you for recognizing our work is pioneering work for crystal property prediction tasks without structural information! We apologize for any confusion or ambiguity present in the initial submission. We hope our response has clarified the contents of the paper, and we are more than willing to engage in further discussion regarding any aspects that remain unclear.

---

> ### Author Response · Authors · 2023-11-21
>
> We kindly ask the reviewer to consider our response that addresses the issues regarding the limited technical novelty and practicality. Should the reviewer have any more concerns, we encourage them to raise these with us. We are more than willing to address any additional questions or concerns the reviewer might have.

---

### Official Review · Reviewer_sAC5 · 2023-10-30

**Soundness:** 2 fair
**Presentation:** 2 fair
**Contribution:** 1 poor
**Rating:** 3
**Confidence:** 5

**Summary:**

The paper attempted to generate a probabilistic representation of stoichiometry from structural information of crystalline materials.

**Strengths:**

The authors should be highly praised for studying important real objects such as solid crystalline materials.

The paper is generally well-written and contains enough details that helped understand the difficulties.

**Weaknesses:**

The keyword "stoichiometry" was not explained but the authors likely meant the chemical composition of a given material, more exactly ratios of weights of involved elements.

The question “Is it possible for stoichiometry-based models to also capture the structural information of crystals?” has the trivial answer "no". Both diamond and graphite in Fig. 1c only a few lines before the question consist of pure carbon but have different crystal structures and vastly different properties, which is known even to school students.

Atomic types were used as easy attributes in almost all past representations, especially for property prediction. Hence there is no need to separately talk about stoichiometry.

The word "problem" appears once (in the quote below), though a writing a rigorous problem statement might have helped the authors to understand the unresolved challenges.

Quote: "it is still an open problem to construct appropriate descriptions of materials, there is a general agreement on effective descriptors that encompass the following principles "Descriptors should 1) preserve the similarity or difference between two data points (preservativity), 2) be applicable to the entire materials domain of interest (versatility), and 3) be computationally more feasible to generate compared to computing the target property itself (computability)".

Comment. If the problem is to find a complete invariant description of a periodic crystal, crystallographers solved this problem nearly 100 years ago by using Niggli's reduced cell of a lattice and then recording all atoms in so-called standard settings, see the book "TYPIX standardized data and crystal chemical characterization of inorganic structure types" by Parthé et al, which applies to all periodic crystals, not only inorganic.

However, all these standardizations have become obsolete in the new world of big and noisy data because the underlying lattice (not even a unit cell) of any periodic crystal is discontinuous under almost any perturbation, which is obvious already in dimension 1.

For example, the set Z of all integers is nearly identical to a periodic sequence with points 0, 1+ep_1, ..., m+ep_m in the unit cell [0,m+1] for any small ep_1,...,ep_m close to 0, though their minimal periods (or unit cells) 1 and m+1 are arbitrarily different.

This discontinuity was reported for experimental crystals already in 1965, see Lawton SL, Jacobson RA. The reduced cell and its crystallographic applications. Ames Lab., Iowa State Univ. of Science and Tech.

A more recent example from Materials Project shows two nearly identical crystals whose unit cells differ by a factor of (approximately) 2
https://next-gen.materialsproject.org/materials/mp-568619
https://next-gen.materialsproject.org/materials/mp-568656

Moreover, atoms in any material always vibrate above absolute zero temperature, so their positions continuously change. As a result, any crystal structure with fixed atomic coordinates in a database is only a single snapshot of a potentially dynamic object, especially for proteins whose structures are often determined often by crystallization.

Hence the new essential requirement for any (better than the past) representations of crystals is a proved continuity under perturbations of atoms.

Any graph representation of a crystal or a molecule is discontinuous because all chemical bonds are only abstract representations of inter-atomic interactions and depend on numerous thresholds on distances and angles, while atomic nuclei are real physical objects.

Quote: "we propose a novel multi-modal representation learning framework for stoichiometry that incorporates readily available crystal structural information"

Comment. If the problem was to determine stoichometry only from a crystal structure, this problem was resolved by the Crystal Isometry Principle, which (briefly) says that any real periodic crystal (with all atomic types, hence stoichometry) is uniquely determined (without any uncertainty) by the geometric structure of atoms (without chemical elements).

See Widdowson et al (NeurIPS 2022) for theoretical proofs and 200+ billion comparisons on the world's largest collection of materials (the Cambridge Structural Database). The underlying invariants have a near-linear time in the motif size and were used for property predictions by Ropers et all (DAMDID 2022) and by Balasingham et al (arxiv:2212.11246).

Since atomic coordinates have continuous real values, the space of materials is continuously infinite. Hence any sixteen (16 million or any large number of) finite datasets cover an infinitely tiny subspace of measure 0 in the full representation space.

**Questions:**

What are the reasons to invent a new word such as "preservativity" in the first introductory paragraph without giving a definition, while the concepts of invariants (https://en.wikipedia.org/wiki/Invariant_(mathematics)), metric (https://en.wikipedia.org/wiki/Metric_space) and continuity (https://en.wikipedia.org/wiki/Continuous_function) have been used for centuries?

On what probabilistic space is the random variable below defined?

Quote: "we propose Polymorphic Stoichiometry Representation Learning (PolySRL), which
aims to learn the representation of stoichiometry as a random variable of polymorphs instead of a single deterministic representation"

Quote: "assuming that polymorphs with an identical stoichiometry follow the same Gaussian distribution, PolySRL models each stoichiometry as a parameterized Gaussian distribution"

Is it a realistic assumption that all existing polymorphs of any fixed stoichiometry can be synthesized with probability proportional to a Gaussian distribution? Could you please specify this distribution in the case of pure carbon, including diamond and graphite?

How many CPU hours and hidden parameters were used in the experiments of section 5?

Did the authors know about the classical results in crystallography cited above, starting from Niggli (1927), Lawton (1965), and Parthe (1987)?

---

> ### Author Response · Authors · 2023-11-14
>
> Thank you for recognizing the importance of the problem! We are more than willing to clarify the reviewer’s concerns one by one.
>
> **W1. Stoichiometry was not explained.**
>
> We thank the reviewer for the suggestion that can definitely improve the presentation of our paper! Although we have provided the definition of stoichiometry in the footnote of the first page, we fully agree with the reviewer that its placement may not have been prominent. Therefore, we moved the definition of stoichiometry to the main introduction paragraph (colored in blue)!
>
> **W2. Possibility of stoichiometry-based models to capture the structure information of crystals.**
>
> Thank you for pointing out our research question! As mentioned by the reviewer and depicted in Fig 1 (c) in the main manuscript, it is obvious that stoichiometry itself cannot capture the information about the structural information. However, thanks to the universal approximation theorem [1], we believe that deep neural networks possess the capacity to learn a function that correlates stoichiometry with its corresponding structural information.   Furthermore, a recent application of generative models to deduce stable crystal structures from given stoichiometries [2] provides empirical support that models based on stoichiometry can indeed infer the structural details of crystals.
>
> [1] Multilayer feedforward networks are universal approximators, Neural networks 1989
>
> [2] Crystal Structure Prediction by Joint Equivariant Diffusion, NeurIPS 2023
>
> **W3. Atomic types and stoichiometry were used as easy attributes in almost all past representations.**
>
> We fully agree that atomic types and stoichiometry have been widely employed in numerous preceding studies. We have recognized these prior contributions in our research and references to the relevant works are comprehensively listed in Section 2.2 of the main manuscript.
>
> **W4. Problem Description.**
>
> First of all, we would like to clarify our research problem statement. In materials science, it is a common situation where we cannot observe the exact structural information of crystalline materials during materials synthesis, since the atomic-level rearrangement of the material occurs through mixing of raw materials, heat treatment, and solvent reactions. However, the majority of machine learning models in materials science are built on top of the structural information, restricting their applicability in the real-world material discovery process. Therefore, in this paper, we aim to build a machine learning model that can predict the properties of materials given only stoichiometry.
>
> **W4-1. Find a complete invariance description of a periodic crystal?**
>
> We agree with the reviewer that the challenge of identifying an invariant description of a periodic crystal was addressed almost a century ago, and also recognize that inductive biases related to invariances in the architectural design of deep neural networks for crystalline materials [1]. However, in this paper, we focus on learning the probabilistic representation of stoichiometry where the crystal structure remains unidentified, as opposed to seeking a comprehensive invariant description of a periodic crystal.
>
> [1] Periodic graph transformers for crystal material property prediction, NeurIPS 2022
>
> **W4-2. Determining stoichiometry only from a crystal structure?**
>
> We concur with the reviewer that, in theory, stoichiometry is uniquely determined by the geometric arrangement of atoms, as also described in our response to [reviewer rRHK](https://openreview.net/forum?id=yuGqe96t9O&noteId=D5aw44Ub3h).
> However, in this paper, our objective is not to determine stoichiometry itself, but to learn the encoder that establishes a representation of stoichiometry where the crystal structure is not known.
>
> **Q1. Reasons to invent a new word.**
>
> Thank you for your suggestion! We originally coined the term “preservativity” to denote the retention of similarities or differences among data points within the latent space. Upon the reviewer’s suggestion, we agree that “invariance” is a suitable alternative and have accordingly replaced “preservativity” with “invariance” throughout the text (colored in blue).
>
>
> **Q2. What probabilistic space defined?**
>
> The random variable is defined on the probability space whose measure is Gaussian distribution parameterized with deep neural networks with $\mathbb{R}^F$ dimensional space as sample space.
>
> On the other hand, acknowledging that the term “random variable” may cause confusion for readers, we have decided to revise the sentence as follows: we propose Polymorphic Stoichiometry Representation Learning (PolySRL), which aims to learn the representation of stoichiometry as a probabilistic representation of polymorphs instead of a single deterministic representation.

---

> ### Author Response · Authors · 2023-11-14
>
> **Q3. Appropriateness of Gaussian assumption.**
>
> We appreciate the reviewer for providing a great point for discussion! We have incorporated this discussion into the discussion section located in **Appendix H**.
>
> In fact, the particles that make up a material are distributed discretely [1], and the material itself manifests following a distinct probability distribution, which is a composite of the discrete distributions attributed to its constituent particles [2]. However, parameterizing such discrete distributions for each material is impossible, and therefore, an alternative distribution that can approximate the actual distribution is required.
>
> Therefore, we choose the Gaussian distribution as an approximate, which has multiple advantages when incorporated with deep neural networks as follows:
> - Efficient gradient computation is available with a reparameterization trick.
> - Analytical computation of KL divergence is available and theoretically guaranteed.
>
> Furthermore, we note in Section 5.3 that the uncertainty, which is obtained from the variance of the Gaussian distribution, is consistent with established materials science expertise. Given that our approach involves a probabilistic representation of stoichiometry to offer both precise representation and model uncertainty, we believe that the Gaussian distribution is effectively fulfilling its intended role.
>
> [1] Density functional theory of electronic structure, The journal of physical chemistry 1996
>
> [2] Exploiting solid-state dynamic nuclear polarization NMR spectroscopy to establish the spatial distribution of polymorphic phases in a solid material, Chemical Science 2023
>
>
> **Q4. How many CPU hours and hidden parameters?**
>
> Since we have trained our model on GPU, we will report the GPU hours used for this process. It takes 3 hours to train the model 100 epochs with the MP dataset which consists of 80,162 distinct stoichiometries with their corresponding 112,183 crystal structures in 24 GB NVIDIA GeForce RTX 3090. Moreover, We have 1,445,800 hidden parameters for the stoichiometry encoder and 2,541,811 hidden parameters for the structural encoder.
>
>
> **Q5. Did the authors know about the classical results in the crystallograph cited above?**
>
> While we recognize the fact that 1) there exists an invariant description of a periodic crystal, and 2) the stoichiometry is uniquely determined by the geometric arrangement of atoms, we were not fully aware of the specific details within the cited papers. We greatly appreciate the seminal research that has laid the groundwork for current advancements in the field, and we will ensure to acknowledge this body of work appropriately.
>
> **Final Remark.**
>
> Once again, we appreciate the reviewer highlighting valuable discussion points in our study. We also offer our apologies for any confusion or ambiguity in our initial submission. We hope that our response has made the details of the paper clearer, and we are fully prepared to continue the conversation about any points that may still need further explanation.

---

> > ### Comment · Reviewer_sAC5 · 2023-11-19
> >
> > Thank you for the answers and references.
> >
> > > it is obvious that stoichiometry itself cannot capture the information about the structural information. However, thanks to the universal approximation theorem [1], we believe that deep neural networks possess the capacity to learn a function that correlates stoichiometry with its corresponding structural information.
> >
> > If "it is obvious that stoichiometry itself cannot capture the information about the structural information", then how can you "believe that deep neural networks possess the capacity to learn a function that correlates stoichiometry with its corresponding structural information" and what is the exact meaning of "correlates" here?
> >
> > >[2] provides empirical support that models based on stoichiometry can indeed infer the structural details of crystals.
> >
> > If the reference was to https://arxiv.org/abs/2309.04475, this paper also ignores the discontinuity of Niggli's reduced cell under almost any tiny permutation of atoms.
> >
> > >we aim to build a machine learning model that can predict the properties of materials given only stoichiometry.
> >
> > If the given stoichiometry is pure carbon, how can vastly different properties of diamond and graphite be predicted from the same input?
> >
> > >we focus on learning the probabilistic representation of stoichiometry where the crystal structure remains unidentified
> >
> > What is the space on which probability distributions are considered?
> >
> > >learn the encoder that establishes a representation of stoichiometry where the crystal structure is not known
> >
> > Does "representation" here mean a geometric structure of a crystal?
> >
> > >Analytical computation of KL divergence is available
> >
> > If KL referred to https://en.wikipedia.org/wiki/Kullback-Leibler_divergence, this function fails the metric axioms.
> >
> > >We have 1,445,800 hidden parameters for the stoichiometry encoder and 2,541,811 hidden parameters for the structural encoder.
> >
> > If a training set changes, should these 4 million parameters be recomputed?

---

> ### Author Response · Authors · 2023-11-20
>
> Thank you for your time and efforts in reviewing our paper! We are happy to discuss more about our paper.
>
> > If "it is obvious that stoichiometry itself cannot capture the information about the structural information", then how can you "believe that deep neural networks possess the capacity to learn a function that correlates stoichiometry with its corresponding structural information" and what is the exact meaning of "correlates" here?
>
> Firstly, by "correlates" we are referring to the mapping between stoichiometry and structural information, implying that a neural network can be trained to map stoichiometry with its corresponding structural details.
>
> Although it's clear that stoichiometry alone doesn't encompass structural information, our premise is based on the existence of a function that can relate stoichiometry to its specific crystal structure. Thanks to the universal approximation theorem, this allows us to train deep neural networks to approximate this function using pairs of stoichiometry and their associated structures.
>
> > If the reference was to https://arxiv.org/abs/2309.04475, this paper also ignores the discontinuity of Niggli's reduced cell under almost any tiny permutation of atoms.
>
> Although this paper does not consider the discontinuity of Niggli’s reduced cell, predicting crystal structure based on stoichiometry demonstrates that deep neural networks can learn the mapping function between stoichiometry and its possible stable structure.
>
> We kindly request the reviewer to acknowledge the recent advances of deep generative models for crystal material generation, which sheds light on the possibility of deep neural networks to learn the crystal structure [1, 2 ,3].
>
> [1] Crystal Diffusion Variational Autoencoder for Periodic Material Generation, ICLR 2022.
>
> [2] Crystal Structure Prediction by Joint Equivariant Diffusion, NeurIPS 2023.
>
> [3] Towards Symmetry-Aware Generation of Periodic Materials, NeurIPS 2023.
>
> > If the given stoichiometry is pure carbon, how can vastly different properties of diamond and graphite be predicted from the same input?
>
> As the reviewer pointed out, a model relying solely on stoichiometry may not be able to distinguish vastly different properties of materials with differing structures. To overcome the limitation, our model accounts for uncertainties in the materials, and the ability to quantify these uncertainties is crucial in the real-world application of machine learning, especially for virtual screening in scientific discovery [1, 2].
>
> Additionally, it's common in wet lab experiments to have only stoichiometry information available as we responded to reviewer [jsDD](https://openreview.net/forum?id=yuGqe96t9O&noteId=efaOs4V5j0). In such cases, models that depend on structural information become inapplicable, presenting a challenge for their use in materials discovery. Consequently, while stoichiometry-based models may not differentiate between materials with the same stoichiometry but different structures, we propose that they represent a promising approach for machine learning to make significant contributions in the field of materials science.
>
> [1] Machine learning framework for enhanced molecular screening. Chemical science 2021.
>
> [2] Accelerating high-throughput virtual screening through molecular pool-based active learning. Chemical science 2021.
>
> > What is the space on which probability distributions are considered?
>
> This space is the embedding space of stoichiometry, which also is the output space of the stoichiometry encoder.
>
> > Does "representation" here mean a geometric structure of a crystal?
>
> In this context, "representation" refers to the vector output produced by the stoichiometry encoder, which transforms stoichiometry into a vector space. This is distinct from the geometric structure of a crystal.
> By contrasting this representation with the representation from “structural encoder”, representation of stoichiometry can contain the structural information of the material.

---

> ### Author Response · Authors · 2023-11-20
>
> > If KL referred to https://en.wikipedia.org/wiki/Kullback-Leibler_divergence, this function fails the metric axioms.
>
> The reviewer correctly noted that KL divergence does not adhere to the metric axiom due to its lack of symmetry and failure to fulfill the triangular inequality. Nevertheless, within the machine learning community, KL divergence is often considered the most intuitive approach for comparing two probability distributions [1, 2]. This is because, as a core concept in information theory, it effectively measures the closeness of two probability distributions in terms of bits. Therefore, we decided to use KL divergence to measure the difference between the learned distribution and prior distribution.
>
> [1] Deep Variational Information Bottleneck, 2017 ICLR
>
> [2] Auto-Encoding Variational Bayes, 2014 ICLR
>
> > If a training set changes, should these 4 million parameters be recomputed?
>
> If the training set undergoes complete change, it would be necessary to train all parameters from the beginning. However, in practical scenarios, such a situation is unlikely since our model is trained with all the possible stoichiometry-structure pairs in Materials Project website. Thus, in real-world applications, if a new combination of stoichiometry and crystal structure emerges, we can efficiently update the model by training it with the new data, rather than starting from scratch.

---

### Official Review · Reviewer_rRHK · 2023-11-01

**Soundness:** 3 good
**Presentation:** 3 good
**Contribution:** 3 good
**Rating:** 8
**Confidence:** 4

**Summary:**

This paper tackles the problem of learning representations of stoichiometries (ratio of chemical elements in a compound) for materials property prediction. In particular, this papers addresses the challenge of polymorphism, which refers to the fact that a single stoichiometry can correspond to multiple, diverse materials due to the different ways in which atoms can be arranged to form thermodynamically stable structures. The paper presents a machine learning algorithm where the structural information of compounds available in a data set is used to train a probabilistic embedding of the stoichiometry via a contrastive loss. The model optimises the parameters of a multivariate Gaussian distribution such that the graph representations of a stoichiometry and the structural graph representations of its polymorphs is minimised, while maximising the distance to other structures. The evaluation includes various data sets focused on different material properties and comparisons with a diverse set of methods.

**Strengths:**

I have enjoyed reading this paper, it is interesting and well written and generally the strengths of the paper outweigh my concerns, which I mention in the next section. I will highlight here some of these strengths.

First of all, the paper identifies an important challenge in the application of machine learning methods for materials discovery. Namely, that the structural information of new materials is not readily available, as the standard way to obtain it depends of computationally expensive simulations, for example with Density Functional Theory (DFT). Therefore, many of the most accurate methods in the machine literature for modelling materials and molecules, which are GNNs that use the structural information as input, cannot be effectively applied to new candidates of which we may only know the composition. This is an important subject that has received relatively little attention and this paper proposes a method that seems to obtain good results on various benchmarks.

The extensive evaluation of the method is also a strength of this paper. The authors compare their method against several diverse baselines, on an array of data sets focused on different materials properties and they include an ablation study of some of the most important components of the algorithm.

Regarding the method itself, I think it is an original and reasonable idea to use a contrastive loss between the structural representation of polymorphs and the stoichiometry representation of their composition. This still leverages the potential of GNNs to learn good structural representations, but allows using only the stoichiometry representation at inference time. I believe there could be multiple variations of the proposed method and I was initially skeptical of some of the specific details of the method, but the results seem to support the choices, and future work may explore other alternatives.

Finally, the paper is for the most part well written and easy to follow - with some exceptions I discuss below.

**Weaknesses:**

I would encourage the authors to improve the clarity of the last part of the paper, namely Section 5. While the quality is not poor, I did notice a decrease in the quality and clarity of this section with respect to the methods part of the paper, which I found very clear. Note that the first paragraph of 5.1 contains a few typos or grammar mistakes. Also, the size of the figures in this section are too small.

An exception of this clarity in the methods section is Section 4.3, where I believe the notation is a bit confusing. For instance, $\mathcal{P}^b$ is used here for the first time to refer to what I believe is the structural graph, previously referred to as $\mathcal{G}^b$. Also, should the superscript not be $b$ instead of $a$ because it refers to the structural graph? In fact, the superscript of $z_p$ is $b$. Or perhaps I am missing something. In this section as well, $m$ is used in the conditional probability without defining it.

Although there is additional information in the appendix, I would suggest to include some details regarding the data splits (train, validation, cross-validation) in the main paper as well to describe the evaluation protocol.

Machine learning models that predict the structure given the stoichiometry can be reasonable or even better alternatives to richer representations of the stoichiometry as proposed here. I would have appreciated to find a discussion of the pros and cons of these two approaches and ideally a review of crystal structure prediction methods.

**Questions:**

Some disconnected questions or comments:

- Why the figure of merit metric is not included in the Table with transfer learning results?
- Several times throughout the paper it is mentioned that there exists a one-to-many relationship between the stoichiometry and the possible structures. Does the structural graph contain information about the atoms species? I wonder if it is also the case, otherwise, that one structures actually corresponds to more than one stoichiometry.
- Regarding the "collapsed dimensions" of the representation, I wonder whether it is actually due to the fact that some stoichiometries have no known polymorphs in the data sets and therefore their variance should in fact be zero. What do you think?

---

> ### Author Response · Authors · 2023-11-14
>
> Thank you for acknowledging the importance of the research topic and our efforts in extensive experiments! We are more than willing to clarify the reviewer’s concerns one by one.
>
> **W1 & W3. Improving the clarity of Section 5 / Including some details regarding data splits.**
>
> We greatly appreciate the reviewer for pointing out the unclarified parts in our paper, as this will undoubtedly enhance the presentation of our research. In response to the reviewer's feedback, we have implemented the following revisions, which are highlighted in blue in the main manuscript:
>
> - We have corrected the grammatical errors and typographical mistakes in the first paragraph of section 5.1
> - We have increased the size of Figure 3 on Page 8
> - We have increased the font size of Figure 4 on Page 9
> - We have increased the size of Figure 5 on Page 9
> - We have included more details on data splits in the third paragraph of Section 5.1
>
> Additionally, we will ensure a thorough review of our paper to correct any grammatical errors throughout the document.
>
> **W2. Confusing notation in Section 4.3.**
>
> First of all, we apologize for any confusion caused by the use of the terms $\mathcal{P}^a$ and $m$.
> In fact, $\mathcal{P}^a$ is intended to represent the set of polymorphic crystal structures associated with the stoichiometry graph $\mathcal{G}^a$, as indicated in line 10 of Section 3.2. However, we fully agree with the reviewers that this notation might lead to ambiguity since superscript $b$ is consistently used for the structural graph. Therefore, to clarify our notation, we have decided to change the term $\mathcal{P}^a$  into $\mathcal{P}^{\mathcal{G}^a}$.
> Moreover, we apologize for missing the precise definition of $m$, which is the indicator function of value 1 if $\mathcal{P}^{\mathcal{G}^a}$ is the set of polymorphic structures corresponding to $\mathcal{G}^a$ and 0 otherwise. We highlighted the modified term in blue color in the main manuscript.
>
> **W4-1. Discussions on the pros and cons of using stoichiometry compared to crystal structure prediction (CSP).**
>
> Thank you for pointing out a great discussion point! In our opinion, here are the advantages and disadvantages of employing stoichiometry-based models versus crystal structure prediction (CSP) methods.
>
> **Advantages of stoichiometry-based models**
> - CSP methods firstly predict the structure of crystals, and then train another model for downstream tasks based on the predicted structure. Therefore, accurate crystal structure prediction is crucial, as the success of downstream tasks hinges on the initial accuracy of crystal structure prediction. However, it is also well-known that CSP methods encounter difficulties with complex systems, potentially hindering their efficacy in various downstream applications [1]. On the other hand, since stoichiometry-based models derive predictions directly from the elemental composition, they avoid the pitfalls associated with predicting the precise structure.
> - Furthermore, we argue that models based on stoichiometry could be applicable to a broader range of materials, including doped and alloyed materials. Doping and alloying involve transformative processes that introduce a level of disorder into a conventional structure, making the precise prediction of the resulting material's structure exceedingly challenging for CSP methods [2]. On the other hand, stoichiometry-based models, which rely solely on the ratio of constituent atoms, offer greater versatility across a diverse spectrum of materials.
> - Additionally, the computational expense of stoichiometry-based models is expected to be significantly lower than that of CSP models. Traditional CSP methods, particularly those reliant on Density Functional Theory (DFT), are known to be computationally demanding [3]. On the other hand, it is acknowledged that stoichiometry-based approaches circumvent the need for extensive computational resources, thereby offering a cost-effective alternative [4].
>
> **Disadvantages of stoichiometry-based models**
> - Given that the crystal structure is a key determinant of a material's properties, CSP models that can precisely predict crystal structure have the potential to surpass the performance of stoichiometry-based models.
>
> [1] MLatticeABC: generic lattice constant prediction of crystal materials using machine learning. ACS omega 2021.
>
> [2] Effects of doping on the crystal structure of poly (3-alkylthiophene), Journal of Materials Chemistry 2.9 1992.
>
> [3] Density functional theory: a practical introduction, John Wiley & Sons 2022.
>
> [4] Representations of materials for machine learning, Annual Review of Materials Research 53 2023.

---

> ### Author Response · Authors · 2023-11-14
>
> **W4-2. A review of crystal structure prediction methods.**
>
> While we have examined the pros and cons of employing stoichiometry-based models as outlined in W4-1, we acknowledge the critical significance of crystal structure prediction (CSP), which has been a long-standing problem in physical sciences [1]. We fully agree with the reviewer on the significant relevance of CSP to our work, and we decided to add the following review in **Appendix G**!
>
> Crystal structure prediction (CSP) is the process of determining the stable three-dimensional structure of a compound from its chemical composition alone. Conventional CSP methods often combine density functional theory (DFT) with optimization algorithms. These algorithms carry out an iterative process searching for stable states that align with the local energy minima, while DFT is utilized to assess the energy at each step of the iteration [2, 3, 4]. A recent development in the field is DiffCSP [5], which employs a deep generative model within a diffusion framework to simultaneously optimize lattice matrices and atomic coordinates, offering a novel approach to the CSP problem.
>
> It's important to highlight the growing interest among researchers in generating crystal structures without predefined stoichiometry, a direction distinct from CSP. The groundbreaking CDVAE [6] integrates a variational autoencoder (VAE) architecture with a diffusion-based decoder to produce the types of atoms, their coordinates, and lattice parameters. Unlike CSP-focused methods, CDVAE's primary goal is the generation of random crystal structures, providing a different avenue in the field of crystallography research.
>
> [1] Cryptic crystallography, Nature Materials 2022
>
> [2] Ab initio random structure searching, Journal of Physics: Condensed Matter 2011
>
> [3] Crystal structure prediction accelerated by bayesian optimization, Physical Review Materials 2018
>
> [4] Structure prediction drives materials discovery, Nature Reviews Materials 2019
>
> [5] Crystal Structure Prediction by Joint Equivariant Diffusion, NeurIPS 2023
>
> [6] Crystal diffusion variational autoencoder for periodic material generation, ICLR 2022
>
>
> **Q1. Figure of merit for Transfer Learning.**
>
> Space limitations in our submission led us to exclude some results, which may result in a lack of clarity in our presentation. As done in representation learning, we determined the figure of merit using a rule-based approach to evaluate model predictions. Our observations indicate that models incorporating 3D information significantly surpass those that do not take 3D data into account.
>
> |             | 300K          | 600K          |
> |-------------|---------------|---------------|
> | Rand Init   | 0.073 (0.024) | 0.218 (0.076) |
> | GraphCL     | 0.064 (0.014) | 0.202 (0.086) |
> | MP Band G.  | 0.067 (0.011) | 0.206 (0.119) |
> | MP Form. E. | 0.069 (0.010) | 0.223 (0.097) |
> | 3D Infomax  | 0.066 (0.010) | 0.154 (0.032) |
> | PolySRL     | 0.058 (0.012) | 0.179 (0.052) |
>
>
> **Q2-1. Structural graph contains information about the atom species?**
>
> No, the structural graph in PolySRL does not include atom species information in the structural graph. Instead, as detailed in line 11 of Section 3.2, each node in the structural graph is associated with a learnable feature vector to make sure we utilize only structural information. Traditional machine learning methods that use structural graphs [1, 2] typically embed atom species information within the nodes of the graph. However, we intentionally omitted atom species information to ensure that our focus was solely on structural information. This methodology has also been validated in accurately estimating the mutual information between 2D molecule graphs and their corresponding 3D point clouds [3].
>
> [1] Crystal graph convolutional neural networks for an accurate and interpretable prediction of material properties, Physical review letters 2018
>
> [2] Atomistic line graph neural network for improved materials property predictions, npj Computational Materials 2021
>
> [3] 3D Infomax improves GNNs for Molecular Property Prediction, ICML 2022

---

> ### Author Response · Authors · 2023-11-14
>
> **Q2-2. One structure corresponds to more than one stoichiometry?**
>
> Yes, this can also be the case since we excluded the atom species information in the structural graph. That is, it is possible that more than one stoichiometry can have the same structural graphs. However, we believe this is due to the design choice of structural graph,  not due to the knowledge in materials science. Theoretically, a structural graph that assigns atom species to each node for machine learning should correspond to a unique stoichiometry.
>
> On the other hand, an interesting phenomenon where a single structure corresponds to multiple stoichiometries has indeed been observed by scientists in real-world scenarios. However, this is not a widespread occurrence and is typically examined at a case study level within the field of chemistry [1, 2].
>
> [1] Multi-stoichiometric quasi-two-dimensional W n O 3n− 1 tungsten oxides, Nanoscale 2020.
>
> [2] Crystal structures capture multiple stoichiometric states of an aqueous self-assembling oligourea foldamer, Chemical Communications 2021.
>
>
> **Q3. Regarding collapsed dimensions.**
>
> Yes, we believe “collapsed dimension” is due to the fact that some stoichiometries have no known polymorphs in the data sets, and we posit that Figure 4 corroborates this understanding. As depicted in Figure 4, the predicted uncertainties for stoichiometries associated with a single possible structure (that is, those without known polymorphs in the dataset) exhibit the lowest levels of uncertainty compared to other groups. This suggests that stoichiometries with a single possible structure are likely to present more collapsed dimensions.
>
> **Final Remark.**
>
> Once again, thank you for strongly supporting our work, and we apologize for any confusion or lack of clarity in our initial submission. We hope our response has clarified the contents of the paper, and we are more than willing to engage in further discussion regarding any aspects that remain unclear.

---

> > ### Comment · Reviewer_rRHK · 2023-11-21
> > **Response to authors**
> >
> > Thank you for the detailed answer to my review. I appreciate the discussion of my comments and the decisions to include part of these in the manuscript. I also welcome to decisions to clarify the notation.
> >
> > I think the decision of not including atom species in the structural graph could be defended. However, since the exclusion of atom species from the structural graph yields in theory a many-to-many scenario regarding stoichiometries and structures, would it not be more appropriate to discuss this explicitly in the paper? Currently, the text follows the premise of a one-to-many scenario.
> >
> > Regarding collapsed dimensions, after your confirmation that they will likely correspond to to stoichiometries with no known polymorphs, then I conclude that this is not a phenomenon to avoid, but rather a natural consequence of the modelling design. However, the paper treats collapsed dimensions as an indicator of "worse" fit. I wonder whether it would be appropriate to either tune the analysis regarding collapsed dimensions, or discuss this as a potential limitation of the method.

---

> > > ### Author Response · Authors · 2023-11-21
> > >
> > > We are pleased to hear that you are satisfied with our revised manuscript and appreciate your additional suggestions!
> > >
> > > > Discussion on many-to-many relationship between stoichiometry and structure.
> > >
> > > We concur with the reviewer that a detailed discussion of this relationship would enhance our paper. Accordingly, we have included our discourse on this topic in Appendix H.
> > >
> > > > I wonder whether it would be appropriate to either tune the analysis regarding collapsed dimensions, or discuss this as a potential limitation of the method.
> > >
> > > Firstly, we apologize for any confusion caused by the discussion of collapsed dimensions in Section 5.2. Regarding the "fitness" of the representation, our observations suggest that the model typically shows improved performance with collapsed dimensions, as illustrated in Figure 3. However, from an "analysis" perspective, this collapsed dimension fails to provide meaningful insights into uncertainty, potentially limiting the practical utility of PolySRL, rather than “worse fit” of the model. Essentially, there seems to be a trade-off in our model between performance and suitability in analysis, which can be modulated by the $\beta$ hyperparameter.
> > >
> > > A notable limitation of our approach is the necessity for manual tuning of the $\beta$ hyperparameter, which requires a process of trial and error to optimize. We have acknowledged this limitation in Section 5.2 of our paper.

---

### Meta-Review · Area_Chair_Lzko · 2023-12-10

**Metareview:**

There are three reviews with rather different scores/comments. One reviewer has a very positive impression of this paper, and the main arguments seem to be the extensive evaluation experiments including many known benchmark datasets and the conceptual idea of using a contrastive loss between structural representations and stoichiometry-based representations. On the other extreme, there is one very negative review, essentially questioning the whole idea of using ML methods for this purpose. Between these extremes, there is a third reviewer who mentions several critical points,  the most fundamental of which is the general question about the possibility to predict details of the structure given only the stoichiometry.  After going again over the the paper, the rebuttal and the discussions, I come to the conclusion that the weaknesses of this paper outweigh the positive aspects. In particular, I am not convinced that the authors could address fundamental questions about the possibility to predict details of the the structure given only the stoichiometry. Although I am convinced that it is possible to lean a function that correlates stoichiometry with some  structural aspects, I am less convinced that it is possible to learn too many structural details on the basis of just a few examples for each stoichiometry. And I was a bit irritated by the authors' comments about the role of universal approximation theorems for neural networks in this context:  I would say that the criticism does not concern the potentially missing flexibility of a network, but rather the general difficulty of predicting structural details based on the stoichiometry alone. In summary, I had the impression that this is a typical borderline paper, which has several positive aspects, but also several weaknesses. I think that the weaknesses address rather deep conceptual questions, and I am not convinced that the authors could addressed these questions in a convincing way during the rebuttal period. For me, there are still too many question marks, and therefore I recommend rejection.

**Justification For Why Not Higher Score:**

There are too many fundamental questions about the main idea...

**Justification For Why Not Lower Score:**

N/A

---

### Decision · Program_Chairs · 2024-01-16

Reject